# The theory of planned behavior as a behavior change model for tobacco control strategies among adolescents in Botswana

Roy Tapera[1]*, Bontle Mbongwe[1], Magen Mhaka-Mutepfa[2], Andrew Lord[3], Nthabiseng A. Phaladze[4], Nicola M. Zetola[5]

1 Deaprtment of Environmental Health, School of Public Health, University of Botswana, Gaborone, Botswana, 2 Department of Psychology, Faculty of Social Sciences, University of Botswana, Gaborone, Botswana, 3 Good Business, London, United Kingdom, 4 School of Nursing, University of Botswana, Gaborone, Botswana, 5 Department of Radiation Oncology, University of Pennsylvania, Philadelphia, Pennsylvania, United States of America

* ztapera@gmail.com

**Data Availability Statement:** All relevant data are within the manuscript and its Supporting Information files.

## Abstract

### Background

Behavioral intentions (motivational factors), attitudes, subjective norm (social pressures), and perceived behavioral control promote or discourage smoking behavior among adolescents.

### Objective

To assess students' behavioral intentions, attitudes, subjective norms and perceived behavioral control on smoking using the Theory of Planned Behavior. The prevalence of smoking among the adolescents is also calculated.

### Methods

In this cross-sectional study, structured self-administered questionnaires were used to collect data from adolescents in primary and secondary schools. Data on demographics, behavioral intentions, attitudes, subjective norms, and perceived behavioral control towards smoking were collected. Pearson product moment correlations and logistic regression models were used to determine factors associated with current smoking.

### Results

A total sample of 2554 (mean age = 15; Range = 12–18 years) students participated in the study. Twenty-nine percent (n = 728) of the students had tried smoking at least once. Smoking was predicted by attitudes, subjective norms, perceived behavioral control and intention. There was a strong association between having a parent or guardian, caregiver or close friend who smoked (p < 0.001) and being a smoker. The majority of students (57%) conveyed that adults talked to them about the harmful effects of cigarette smoking and 50% had discussed smoking concerns with their friends. Students who had positive attitudes towards

**Funding:** This research was commissioned by Good Business (London) as part of a social marketing programme designed to prevent teenage girls from taking up smoking in Botswana. This was funded through a grant from the Bill and Melinda Gates Foundation Grant number OPP1082662. The funder had no role in the study design, collection, analysis and interpretation of the data; writing the report or the decision to submit the report for publication. The views expressed in this paper are therefore those of the authors and do not necessarily reflect the views of the funding body.

**Competing interests:** This research was commissioned by Good Business (London) as part of a social marketing programme designed to prevent teenage girls from taking up smoking in Botswana. The commercial affiliation did not play any role in the study." Please note: This does not alter our adherence to PLOS ONE policies on sharing data and materials.

smoking like "smoking makes you confident" were more likely to be current smokers (OR: 1.63, 95% CI: 1.03–2.59). The feeling or conviction that they could refuse a cigarette if offered was an impediment from smoking (OR: 0.18, 95% CI: 0.13–0.26).

## Conclusions

Attitudes, subjective norms, and perceived behavioral control contributed significantly to the students' smoking. Right attitudes must be cultivated and behavioral control must be strengthened for early effective interventions to curtail smoking among adolescents.

## Background

The World Health Organization (WHO) Report on the Global Tobacco Epidemic (2017) estimates that one in 10 deaths globally is caused by tobacco use. Additionally, about 1.3 billion people in the world currently smoke, and 7 million people die every year from tobacco consumption [1]. Recent research the world over on tobacco smoking revealed that smoking-associated deaths have escalated to 7.2 million lives every year, resulting in more deaths than a combination of HIV/AIDS, malaria and tuberculosis [2], although cigarette smoking is entirely preventable.

Initial exposure for those who end up smoking typically occurs early in adolescence and increases over time [3]. Thus adolescence and early adulthood comprise a critical time for prevention and intervention efforts [4]. It is also well-established that adolescents are more likely to smoke if they have peers or friends who smoke. This association is typically interpreted as evidence of a peer influence effect [5]. Students in the vicinity of smoking peers are also more likely to smoke regardless of being offered a cigarette or not. Passive (imitation) peer influence affects young adult smoking rather than active (pressure) peer influence [6]. Thus, smoking cessation efforts should aim at preventing interaction with smoking peers and advocacy of its impact, particularly in adolescents as they are more impressionable and susceptible to advertising. Additionally, parents or guardians should impart good values, instill positive principles, and model exemplary behavior for their children, a phenomenon known as social modelling [7]. Social modelling may thus be used as a preventative method especially in children.

In a previous study in Botswana, the prevalence of tobacco smoking amongst primary and secondary school students was found to be 10%, whilst 29% reported having tried smoking [3]. In the same study, self-image and acceptance by peers were the strongest predictors of smoking overall (adjusted Odds Ratio [aOR] = 3.13, 95%, Confidence Interval [CI]: 2.67–3.66). The theory of planned behavior (Fig 1) was used to explain smoking behaviors in children in the current study.

The Theory of Planned Behavior (TPB) was developed by Icek Ajzen [17] as an attempt to predict human behavior. TPB provides a framework to identify key behavioral, normative, and control beliefs affecting behaviors. Interventions can then be designed to target and change these beliefs or the value placed on them, thereby affecting attitude, subjective norm, or perceived behavioral control, leading to changes in intentions and behaviors [8,9]. The TPB has also been used successfully to predict and explain a wide range of health behaviors including exercise, smoking and drug use, HIV prevention behaviors, among others [8–16]. Models of behavior, such as the TPB provide a conceptual framework that allows program designers and policy makers to detect the fundamental features that determine behavior and thus design valuable interventions.

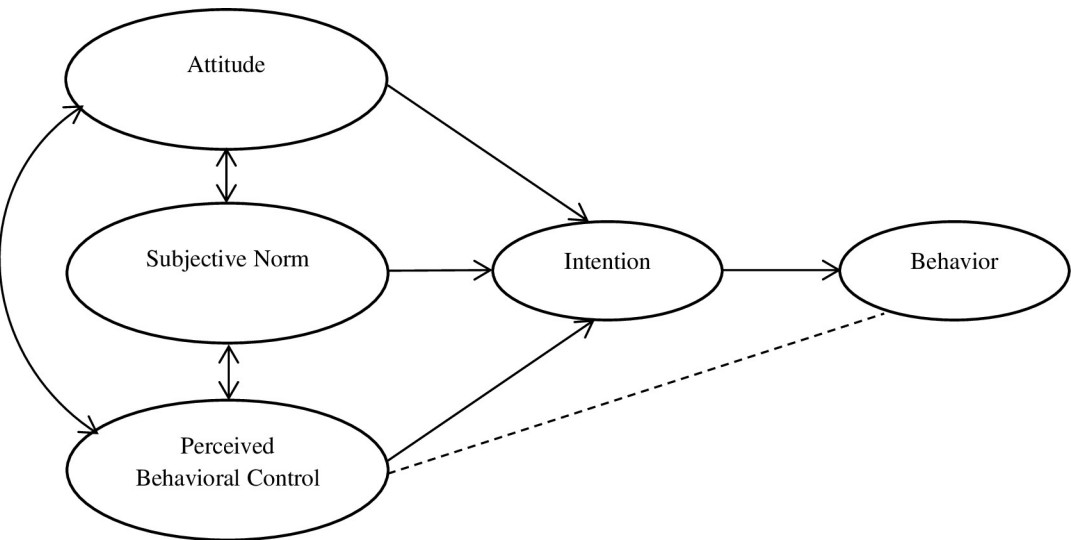

**Fig 1. Theory of Planned Behavior [Adopted from Ajzen, I. [17]].**

The prevention strategies can be tailored if key parental and peer factors (subjective norms) influencing behavior and attitudes of the adolescent towards tobacco smoking are identified. Intentions are also expected to capture the motivational elements that impact behavior as signals of how committed people are prepared and willing to try and apply boundless efforts to perform the behavior [17]. As a general rule, a person with a stronger intention to engage in a behavior is more likely to act as such and perform better than a person who lacks intention. It should be explicit, however, it should be pointed out explicitly that a "behavioral intention" can only be explained if the behavior in question is performed willfully. Although some behavior might meet this requirement, the performance of most individuals may depend on non-motivational factors, such as availability of requisite opportunities and resources (e.g., time, money, skills, synergies) [17].

This article focuses on the second objective of the study done in Botswana [3]. In this study, TPB is utilized to assess attitudes, subjective norms, perceived behavioral control and intentions of adolescents on smoking in Gaborone and Francistown, the two largest cities of Botswana. For example, parental and peer smoking are subjective norms that are strong and significant determinants of the risk of smoking uptake by children and young people in previous studies from 27 European countries, in addition to United States of America and Korea [18–21]. Additionally extant literature shows that intention to smoke is a strong predictor of future smoking [22–24]. Further, perceived behavioral control was found to be a predictor of smoking behavior [25,26]. Perceived behavior control entails an individual's perceptions of his/her ability that includes both internal (e.g., refusal skills) and external (e.g., constraints) behavior. Other previous researchers have found that negative attitudes toward smoking prospectively predict low rates of smoking behavior [27–31] therefore the need to investigate the situation in a sub-Saharan African country like Botswana.

## Materials and methods

### Research design

A cross-sectional survey that assessed attitudes, subjective norms, perceived behavioral control and intentions of adolescents on smoking was done. The cross-sectional study enabled the

prevalence for all factors under investigation in the study to be measured at one point in time. The study was carried out in the two largest districts of Botswana. Data were collected in Gaborone and Francistown primary and secondary schools. The associations among the predictor variables and active smoking were found. The data was collected in January 2014.

**Inclusive and exclusive criteria.** The study population included boys and girls aged 12 to 18 years. For children under 18 years, guardian or parental consent was sought. Children who were not in school at the time of data collection were excluded. Mentally challenged and children whose parents/guardians declined to provide assent or could not give consent did not take part in the study.

## Sample size and sampling

A total of 3000 students from 75 schools consisting of 25 primary, 25 junior secondary, and 25 senior secondary from public and private schools were randomly selected using multistage proportionate sampling. The list of schools was provided by the Ministry of Basic Education. The schools were further stratified according to whether they were private or public. Some seven schools had to be skipped because of bureaucratic challenges.

A sampling fraction was calculated to select participants concerning the population of each school. For each school the students were selected using a systematic random sampling technique from the available class registers.

## Data collection

Self-administered questionnaires were used to collect data. The questionnaire comprised socio-demographic variables, TPB constructs and some questions from the Global Youth Tobacco Survey (GYTS). A brief description of the questionnaire is provided below:

**The main outcome was** "Active smoking", which was defined as having smoked at least 1 cigarette 30 days prior to data collection.

The secondary outcomes were the number of cigarettes smoked in the month before data collection and intention to smoke.

## Main exposures of interest were

*Demographic characteristics*: Three items were included in the questionnaire to elicit personal information on level of study, sex and age; *Attitudes towards smoking*: Four items on attitudes towards smoking derived from GYTS were included (see examples in Table 2); *Subjective Norms*: Three items derived from GYTS were used to assess the influence of adolescent's referent others (parents, guardians and friends) towards their smoking behavior (see examples in Table 2); *Perceived Behavioral Control*: Single item derived from GYTS was used to assess perceived behavioral control to avoid smoking (see examples in Table 2); *Intentions*: Two items derived from GYTS were used to assess adolescent's intention to smoke. For example, it asked "In the next 12 months, do you think you might smoke a cigarette?" or "Do you want to smoke when you grow up?"

## Data analysis

IBM SPSS version 25 (Chicago, IL) statistical software was used for data entry and analysis. Assumptions for normality and homoscedasticity were met as the data was not skewed. Descriptive statistics (frequencies, percentages, and cross tabulation) and Chi-square statistical test was used to determine associations between outcome and exposure variables. Logistic regression was used to establish adjusted odds ratios (AOR) and their 95% confidence intervals (CI), for independent variables (attitudes, subjective norms, perceived behavioral control and intentions)

**Table 1. Demographic characteristics of the study participants.**

|  | Number of respondents (n) | Proportion (%) |
|---|---|---|
| **School Category** | **n = 2550** |  |
| Primary Schools | 594 | 23 |
| Junior Secondary Schools | 1111 | 44 |
| Senior Secondary Schools | 845 | 33 |
| **Gender** | **n = 2432** |  |
| Male | 1021 | 42 |
| Females | 1411 | 58 |
| **Academic level** | **n = 2550** |  |
| Standard 6 | 204 | 8 |
| Standard 7 | 383 | 15 |
| Form 1 | 357 | 14 |
| Form 2 | 383 | 15 |
| Form 3 | 357 | 14 |
| Form 4 | 129 | 5 |
| Form 5 | 537 | 21 |
| Form 6 | 204 | 8 |

Adopted from Mbongwe et al. (2017) [3].

linked with active smoking. For the main analyses, smoking was categorized as a dichotomic variable (yes vs. no). Secondary analyses using linear regression was used to establish the association and smoking was treated as a continuous variable in terms of the number of cigarettes smoked per month. A p-value of less than 0.05 was considered to indicate statistical significance.

## Human subjects

The study was approved by the Ministry of Health and Wellness, the Ministry of Basic Education, University of Botswana Institutional Review Board (IRB) and the Human Research Development

**Table 2. Logistics regression with TPB constructs to predict current cigarette smokers.**

|  | B | Sig | OR | 95% CI for OR | |
|---|---|---|---|---|---|
|  |  |  |  | Lower | Upper |
| **Attitudes** |  |  |  |  |  |
| Positive attitudes towards smoking like "smoking makes you confident" | 0.72 | 0.001 | 1.63 | 1.03 | 2.59 |
| Smoking cigarettes is enjoyable | 0.80 | 0.001 | 2.3 | 1.52 | 3.48 |
| Positive attitudes towards none smokers like "there are cool people who do not smoke" | - 0.36 | 0.003 | 0.54 | 0.31 | 0.95 |
| I think someone my age who does not smoke cigarettes looks well kempt | -0.66 | 0.002 | 0.56 | 0.38 | 0.83 |
| **Subjective Norms** |  |  |  |  |  |
| Participants who had mothers/female guardians or caregivers who smoked. | 1.01 | 0.001 | 2.7 | 1.59 | 4.59 |
| Fathers/male guardians who smoke | 0.50 | 0.001 | 1.7 | 1.4 | 2.1 |
| Students who perceived norms conformity with smoking | 0.90 | 0.001 | 1.3 | 1.10 | 1.57[a] |
| **Perceived Control** |  |  |  |  |  |
| The feeling or conviction that they could refuse a cigarette if a friend offered | -0.86 | 0.001 | 0.18 | 0.13 | 0.26 |
| **Intentions** |  |  |  |  |  |
| The students who intended to smoke or continue to smoke | 0.44 | 0.001 | 1.8 | 1.67 | 2.11[a] |
| **Constant** | -2.22 | 0.001 | 0.11 |  |  |

OR = Odds ratio, CI = confidence interval. Data with [a] is from [3].

Committee of Botswana. All participants agreed to participate. Written consent was sought directly from students who were 18 years. Written consent for participation of persons aged 17 years and younger was sought from their legal guardians and written agreement from the minor.

## Results

### Demographic characteristics of study participants

Forty-four percent (n = 1111) of respondents were from junior secondary schools, whilst 33% (n = 845) and 23% (n = 598) were from senior secondary and primary schools respectively (see Table 1). Out of the 2,550 respondents, 2,432 respondents (95.2%) indicated their gender; 58% were female while 42% were male. The average age of the participants was 15 years with an age range of 12–18 years. Academic levels for participants are also presented in Table 1, where most of the participants were in form 5 (21%).

Current smokers were 261 (10%) and a significant proportion of respondents (29%) had tried smoking cigarettes or any form of tobacco [3].

### Dimension of TPB constructs in association with current cigarettes use

**Intention to smoke.**   Five percent (n = 74) of the respondents compared to 95% (n = 1273) had a conversation with an adult about harmful effects of smoking and have been thinking of smoking in the next 2 months. Three percent (n = 77) of the students had a conversation with an adult and were encouraged to smoke in the last two months. From the 77 students, 16% (n = 12) believed they could smoke in the next 12 months whilst 84% (n = 65) had no intention to smoke (see S1 Table).

S1 Table shows that amongst the students who intended to smoke in the next 12 months (n = 157), few of them 32.5% (n = 51) had a conversation with a friend about the harmful impacts of smoking. Among students with no intention to smoke, the majority 52.8% (n = 1148) had a conversation in the past two months about the harmful effects of smoking. The students who intended to smoke or continue to smoke was calculated [3] and the adolescents had 1.8 times odds of smoking compared to those who had no intention to smoke (aOR = 1.81, 95% CI: 1.67–2.11; refer to Table 2).

**Subjective norms.**   The majority (57%) of students said adults talked to them about the harmful effects of cigarette smoking and 50% had discussed smoking issues with their friends. Asked whether in the past 2 months they had a conversation with any of their friends about smoking; 50% had talked about how harmful smoking is, 33% talked about their feelings toward smoking and 29% talked about their refusal of cigarettes. Ten percent indicated that their friends had encouraged them to try smoking whilst 11% indicated their friends had tried to sell them cigarettes.

Females had higher odds of discussing with friends the harms of smoking and talking about how to refuse cigarette smoking compared to their male counterparts (p = 0.013) and (p = 0.045) respectively. There was no difference between gender and friends when discussing how students felt about smoking (p = 0.327) and having tried to sell cigarettes to each other (p = 0.281).

As to whether any of their close family members smoked, 42% of the respondents indicated that someone else in their close family other than the mother, sister, father or brother smoked. More fathers (14%) than mothers (3%) and more brothers 13% than sisters (4%) smoked (see Table 3).

There was no significant difference between school category (p = 0.376) and having a close family member who smokes. No significant differences were observed by gender (p = 0.450) and having close family members who smoked. There was however a strong association between grade and having a close friend who smokes (p<0.001). Respondents in junior

**Table 3. Respondents report on family members who smoke.**

| Close family member who smoke | Frequency | Proportion (%) |
|---|---|---|
| Mother (or female guardian/caregiver) | 79 | 3 |
| Father (or male guardian/caregiver) | 361 | 14 |
| Older sister | 92 | 4 |
| Older brother | 322 | 13 |
| Someone else in my close family | 1069 | 42 |

secondary school had more close family members smoking (46%) compared to respondents in primary schools (31.4%).

Twenty-nine percent of students had tried smoking. Table 2 shows that a strong association was observed between having tried smoking and having a parent, guardian, care giver or close friend who smokes (p < 0.001). Participants who had mothers/female guardians or caregivers who smoked were 2.7 times more likely to have tried any form of tobacco (OR = 2.7, CI: 1.59–4.59) whilst those with fathers/male guardians who smoke were 1.7 times more likely to have tried smoking cigarettes (OR = 1.7, CI: 1.4–2.1). Students who perceived norms conformity with smoking was calculated [3] and the adolescents were 1.3 times likely to be current smokers than those who did not perceive norms conformity with smoking (aOR = 1.31, 95% CI: 1.10–1.57).

**Attitudes.** Table 2 shows that students who had positive attitudes towards smoking like "smoking makes you confident" and "smoking cigarettes is enjoyable" were more likely to be current smokers (OR = 1.63, 95% CI:1.03–2.59) and (OR = 2.3, 95% CI:1.52–3.48) respectively. Negative attitude towards smoking like "smoking cigarettes is expensive" was a deterrent to smoking. Additionally, positive attitudes towards none smokers like "there are cool people who do not smoke" and "I think someone my age who does not smoke cigarettes looks well kempt were found to be protective from smoking (OR = 0.54, 95% CI: 0.31–0.95) and (OR = 0.56, 95% CI: 0.38–0.83) respectively.

**Perceived behavioral control.** Eighty percent (n = 2043) of the students felt they could refuse a cigarette if a friend offered, and those who could refuse were less likely to be current smokers. Table 2 shows that the feeling or conviction that they could refuse a cigarette if a friend offered was protective from being a smoker (OR = 0.18, 95% CI: 0.13–0.26).

Table 4 represents simple Pearson product moment correlations between the TPB variables and other study variables. Favorable behavioral control for smoking was significantly associated with older adolescents, at higher levels of study (e.g., form 5s), who first tried smoking at an older age, (e.g., 17–18 year olds) who had tried smoking several times and for several days in the last month. Those with subjective norms and intentions that were favorable towards

**Table 4. Associations between TPB's constructs and background variables.**

| | Favourable towards smoking | | | |
|---|---|---|---|---|
| | Attitude | Subjective norm | Perceived Control | Intensions |
| Pearson Correlation | | | | |
| Age | 0.038 | -0.058* | 0.075* | -0.113* |
| Grade/Form | 0.019 | -0.046* | 0.056* | -0.119* |
| How many times have you tried to smoke | -0.036 | -0.090* | 0.325* | -0.363* |
| In the last month how many days did you smoke | -0.055* | -0.090* | 0.250* | -0.392* |
| How old were you when you first tried smoking | -0.023* | -0.052* | 0.186* | -0.180* |

The results in the table show Pearson product moment correlations, the * represents significance at an alpha of 0.01.

smoking tended to be younger, had started smoking at a younger age, were in lower levels of study, had tried smoking on fewer occasions and for fewer days in the last month. Although it was not statistically significant, those with favorable attitudes towards smoking tended to be older and were at higher levels of study. Favorable attitudes towards smoking were significantly associated with having started smoking at a young age and smoking for fewer days in the last month (see Table 4).

## Discussion

Results of the current study indicated that a high percentage of students are active smokers (10%) or had tried smoking (29%). These findings are consistent with previous studies from some Asian countries [32–34] and USA [35]. Amongst those who had tried smoking, a strong association between having tried smoking and having a parent, guardian, and/or care giver who smoked was found. These findings are consistent with previous results [19,36], in which guardians and parents who smoked played an influential role in initiating their children to smoking. Parental smoking may exert its influence on adolescent smoking through various mechanisms, including the availability of cigarettes in the home environment, modeling, the internalization of parental smoking norms, and parents' difficulty in enforcing sanctions against smoking when they also smoke [37]. This finding is consistent worldwide [38, 35] as modelling plays a pivotal role in shaping behavior. The television, a source of modelling was also found to influence smoking behaviors uptake in previous studies [3].

Over 50% of the respondents indicated that someone in their close family smoked. The close relatives included fathers, mothers, brothers, sisters and other relatives. Despite a higher prevalence of male guardians who smoke, female guardians exerted a stronger influence over adolescent smoking. Compared to primary schools, respondents in secondary schools had more close family members and friends who smoked. Subjective norms involving close family members and friends was a significant factor for one to smoke. There are negative implications related to these findings. Consistent with previous studies in the USA [35], that adolescents rate of taking up smoking increased as the number of friends who smoked increased due to peer pressure, the current study also showed the significant impact of peer pressure. It was also noted that the majority of the students discuss with adults and half discuss with friends about harmful effects of cigarette smoking. Similar to previous findings [39–41], these discussions with significant others tend to discourage affiliations with substance using adolescents. The assumptions from Social Control Theory [39–41] indicating that parental constraints deter adolescent delinquency are also consistent with the aforementioned finding.

Literature has shown that parental and peer influence plays a part on girls smoking behavior [42]. This finding is consistent with the findings of the current study where respondents who smoked confirmed that someone their age had offered them a cigarette, with a small number indicating that they had been offered a cigarette by an adult in the last two months. These findings indicate that tobacco prevention interventions that aim to influence the behavior of peers, and parents or care givers, could play an important role in tobacco control amongst adolescents as some caregivers influence adolescents to smoke.

A significant number of adolescents reported having conversations with adults (parents or guardians) on the harms of cigarettes. The parent or guardian's involvement in a smoking cessation program is an important step towards reducing the number of adolescents who take up smoking. Compared to low parent involvement, adolescents whose parents were highly involved in smoking cessation were 0.4 times less likely to smoke [35]. However, there is need to pay attention to empowering adolescents to refuse the offers from adults who sell cigarettes or encourage them to smoke. Half of adolescents reported having conversations with their peers

on the harms of tobacco. Nonetheless, many adolescents had their friends selling tobacco products to them or encouraged them to smoke. These findings have implications for programming and policy as they hinge on issues of accessibility of tobacco products to adolescents and raises issues of awareness on the laws governing the sale of tobacco products to and by minors [3]. Over the short term, it is important to involve family members in anti smoking interventions.

In the current study, students who had positive attitudes towards smoking were more likely to be current smokers and negative attitudes towards smoking were a deterrent to smoking. The foregoing findings were similar to those found among secondary school students in China [32]. The implication is that cognitive based therapy (CBT), could be used to change the positive attitudes in adolescents towards smoking. Establishing the right attitudes toward tobacco control in middle school students is advantageous in reducing their smoking rate [43] and cessation.

Majority (80.1%) of the students stated that they could refuse a cigarette if a friend offered. Those who could refuse were less likely to be current smokers. The feeling or conviction that they could refuse a cigarette if a friend offered was protective from being a smoker. This current finding is similar to meta-analysis findings [44,45] in which behavioral control was a strong determinant of smoking. The majority (86%) of students did not have the intention to smoke in the next 12 months which is important for intervention as behavior is shaped by intention. Decades of research show that the strongest determinants of behavior is one's motivation or intention to engage in that behavior [46]. The few students who intended to smoke or continue to smoke were more likely to be current smokers and this was consistent with the findings a study in China [47].

## Limitations

Although the TBP is a good theory in explaining behavior, critics claim that human behavior is much more robust than the four elements (i.e. attitudes, subjective norm, perceived behavioral control, and intention) of TBP. This criticism has led to the inclusion of other related factors [48,49]. Literature shows that affect and emotions can have indirect effects on intentions and behavior independent of the other predictors in the TPB, and that this possibility is not sufficiently accounted for in the TPB [50–52]. Nonetheless, the author Ajzen supported his theory and postulated that TPB does not propose that people are rational or that they behave logically [53–57]. However, the criticism suggests there should be a shift to using the extended TBP in the realm [48,49]. This implies that future research on smoking that use the TBP should extend the TBP theory by investigating affect and emotion, risk perceptions and healthy literacy among others as they all play a pivotal role in uptake of smoking.

The other limitation for this study is that the study was conducted in urban areas of Botswana and targeted adolescents who were in school. The prevalence and associations may be different if adolescents living in rural areas and in the West of the country were included. Botswana has the San (Bushmen), living in the West and this traditional ethnic group is likely to have different findings with regards smoking attitudes, subjective norms, and perceived behavior control. Future studies should include adolescents living in rural villages and in the West (the San) to make comparisons.

## Conclusion

The results show that the TPB plays an important role in predicting smoking, therefore it can be used in designing interventions for smoking cessation and prevention of smoking among adolescence. Adolescents are impressionable thus prevention methods should be put in place early in their lives. When designing intervention, targeting referent others may be the way to

proceed as they have been found to be influential, particularly through modelling because adolescents see them as role models. The correct attitudes towards smoking must be inculcated in both guardians/caregivers and adolescents and behavioral control must be strengthened so that the early interventions are efficacious.

## Supporting information

**S1 Table. Intension to smoke and having had a conversation with an adult about smoking, and they encouraged me to try smoking.**
(DOCX)

**S1 File.**
(XLSX)

## Acknowledgments

The research team acknowledges the support received from the Ministry of Basic Education as well as the University of Botswana Office of Research and Development.

## Author Contributions

**Conceptualization:** Roy Tapera, Bontle Mbongwe, Andrew Lord.

**Data curation:** Bontle Mbongwe, Andrew Lord.

**Formal analysis:** Roy Tapera, Bontle Mbongwe, Magen Mhaka-Mutepfa, Andrew Lord, Nthabiseng A. Phaladze, Nicola M. Zetola.

**Funding acquisition:** Bontle Mbongwe, Andrew Lord.

**Investigation:** Roy Tapera.

**Methodology:** Roy Tapera, Magen Mhaka-Mutepfa, Nthabiseng A. Phaladze, Nicola M. Zetola.

**Project administration:** Bontle Mbongwe.

**Resources:** Roy Tapera, Bontle Mbongwe.

**Software:** Roy Tapera, Andrew Lord, Nicola M. Zetola.

**Supervision:** Roy Tapera, Bontle Mbongwe.

**Validation:** Roy Tapera, Bontle Mbongwe, Magen Mhaka-Mutepfa, Andrew Lord, Nthabiseng A. Phaladze, Nicola M. Zetola.

**Visualization:** Roy Tapera, Bontle Mbongwe, Magen Mhaka-Mutepfa, Andrew Lord, Nthabiseng A. Phaladze, Nicola M. Zetola.

**Writing – original draft:** Roy Tapera, Bontle Mbongwe, Magen Mhaka-Mutepfa, Nicola M. Zetola.

**Writing – review & editing:** Roy Tapera, Bontle Mbongwe, Magen Mhaka-Mutepfa, Andrew Lord, Nthabiseng A. Phaladze, Nicola M. Zetola.

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
