## [Decision Letter · Decision Letter 0]

11 Feb 2020

PONE-D-20-00392

The Theory of Planned Behavior as a behavior change model for tobacco control strategies among adolescents in two cities of Botswana.

PLOS ONE

Dear Dr Tapera,

Thank you for submitting your manuscript to PLOS ONE. After careful consideration, we feel that it has merit but does not fully meet PLOS ONE’s publication criteria as it currently stands. Therefore, we invite you to submit a revised version of the manuscript that addresses the points raised during the review process.

We would appreciate receiving your revised manuscript by Mar 27 2020 11:59PM. To enhance the reproducibility of your results, we recommend that if applicable you deposit your laboratory protocols in protocols.io, where a protocol can be assigned its own identifier (DOI) such that it can be cited independently in the future. For instructions see: http://journals.plos.org/plosone/s/submission-guidelines#loc-laboratory-protocols

We look forward to receiving your revised manuscript.

Kind regards,

Amir H. Pakpour, Ph.D.

Academic Editor

PLOS ONE

Journal Requirements:

2. Please include your tables as part of your main manuscript and remove the individual files. Please note that supplementary tables (should remain/ be uploaded) as separate "supporting information" files

https://psycnet.apa.org/record/2008-17146-000

https://link.springer.com/article/10.1007/s10964-014-0187-7

In your revision ensure you cite all your sources (including your own works), and quote or rephrase any duplicated text outside the methods section. Further consideration is dependent on these concerns being addressed.

"This work was supported by the Bill and Melinda Gates Foundation grant OPP082662. The funders

did not have any additional role in the study design, data collection and analysis, decision to public

or preparation of the manuscript."

5. Thank you for stating the following in the Funding section of your manuscript:

"This study was funded through the financial support of Good Business Ltd."

We note that you received funding from a commercial source: Good Business Ltd.

6. Thank you for stating the following in the Competing Interests section:

"The authors have declared that no competing interest exists."

We note that one or more of the authors are employed by a commercial company: Good Business Ltd.

7. In your Data Availability statement, you have not specified where the minimal data set underlying the results described in your manuscript can be found. PLOS defines a study's minimal data set as the underlying data used to reach the conclusions drawn in the manuscript and any additional data required to replicate the reported study findings in their entirety. All PLOS journals require that the minimal data set be made fully available. For more information about our data policy, please see http://journals.plos.org/plosone/s/data-availability.

8. Please include a separate caption for each figure in your manuscript.

9. Please ensure that you refer to Figure 2 in your text as, if accepted, production will need this reference to link the reader to the figure.

Reviewers' comments:

Reviewer's Responses to Questions

**Comments to the Author**

1. Is the manuscript technically sound, and do the data support the conclusions?

Reviewer #1: No

2. Has the statistical analysis been performed appropriately and rigorously? 

Reviewer #1: I Don't Know

3. Have the authors made all data underlying the findings in their manuscript fully available?

Reviewer #1: Yes

4. Is the manuscript presented in an intelligible fashion and written in standard English?

Reviewer #1: No

5. Review Comments to the Author

Reviewer #1: The manuscript entitled “The theory of planned behavior as a behavior change model for tobacco control strategies among adolescents in Botswana” recruited 2554 adolescents to study whether the theory of planned behavior (TPB) can be a useful theory to examine how far we can understand the behavior of smoking among adolescents inhabiting in Botswana. I applaud for the authors to collect such valuable data on assessing a widely used theory to study a meaningful and important research question. Specifically, the present study has a strong strength in data collection; that is, the authors used a rigorous sampling method to collect a large sample. However, I have following comments for the authors to elevate their work. In addition, I believe that the manuscript needs to be edited by a native English speaker to make it free from grammatical errors (e.g., P5. Our analyses focuses on…)

1. A major and root problem is that the authors do not clearly indicate whether they studied on the smoking behavior. In many descriptions, the authors only talked about smoking intention (e.g., in the Abstract). However, the authors have collected whether the participants are a current smoker and they sometimes reveal the relationship between the TPB elements and a current smoker or not. Therefore, this causes a lot of confusion when I read the manuscript. I cannot know what the authors stand in using the TPB.

2. In the Abstract, the sentence in the Background is unclear. Specifically, it is unclear whose intentions, whose attitudes, whose subjective norms, and whose perceptions that promote/discourage the smoking behavior of an individual. Also, do authors want to say perceived behavioral control when they mention “people’s perceptions”? If yes, then, the authors should not use people’s perceptions to indicate perceived behavioral control; they are different concepts. Following this, the Objective is not in line with what the authors did in the study. Specifically, the authors studied more than “behavioral attitudes” and “intention”, while the authors only mentioned the two factors in the Objective.

3. In the Introduction, the authors said, “The TPB has also been used successfully to predict and explain a wide range of health behaviors including exercise, smoking and drug use, HIV prevention behaviors, among others (Montaño & Kasprzyk, 2002).” I agree. However, please cite more references to support this sentence. Please refer to the following.

Hou, W.-L., Lin, C.-Y., Wang, Y.-M., Tseng, Y.-H., & Shu, B.-C. (2020). Assessing Related Factors of Intention to Perpetrate Dating Violence among University Students Using the Theory of Planned Behavior. International Journal of Environmental Research and Public Health, 17(3), 923.

Fung, X. C. C., Pakpour, A. H., Wu, K.-Y., Fan, C.-W., Lin, C.-Y., Tsang, H. H. W. (2019). Psychosocial variables related to weight-related self-stigma in physical activity among young adults across weight status. International Journal of Environmental Research and Public Health, 17, 64.

Lin, C.-Y., Broström, A., Årestedt, K., Mårtensson, J., Steinke, E. E., & Pakpour, A. H. (2020). Using extended Theory of Planned Behavior to determine factors associated with help-seeking behavior of sexual problems in women with heart failure: A longitudinal study. Journal of Psychosomatic Obstetrics & Gynecology, 41,54-61.

Lin, C.-Y., Broström, A., Nilsen, P., & Pakpour, A. H. (2018). Using extended Theory of Planned Behavior to understand aspirin adherence in pregnant women. Pregnancy Hypertension: An International Journal of Women’s Cardiovascular Health, 12, 84-89.

Lin, C.-Y., Fung, X. C. C., Nikoobakht, M., Burri, A., & Pakpour, A. H. (2017). Using theory of planned behavior incorporated with perceived barriers to explore sexual counseling services delivered by health professionals in individuals suffering from epilepsy. Epilepsy & Behavior, 74, 124-129.

Strong, C., Lin, C.-Y., Jalilolghadr, S., Updegraff, J. A., Broström, A., & Pakpour, A. H. (2018). Sleep hygiene behaviors in Iranian adolescents: an application of the Theory of Planned Behavior. Journal of Sleep Research, 27(1), 23-31.

Lin, C.-Y., Oveisi, S., Burri, A., & Pakpour, A. H. (2017). Theory of Planned Behavior including self-stigma and perceived barriers explain help-seeking behavior for sexual problems in Iranian women suffering from epilepsy. Epilepsy & Behavior, 68, 123-128.

Lin, C.-Y., Updegraff, J. A., & Pakpour, A. H. (2016). The relationship between the theory of planned behavior and medication adherence in patients with epilepsy. Epilepsy & Behavior, 61, 231-236.

4. The Goals of the study section in the Introduction should be rewritten. Specifically, I cannot see the link between the two paragraphs: the authors said “The current study assessed the effect of peer and parental influences on youth smoking” in the first paragraph and “Our analyses focuses on the four belief-based TPB constructs (attitudes, subjective norms,

perceived behavioral control and intentions) because these are most conducive to change with

persuasive messaging in communication campaigns. The main objective of our study was to

identify the key beliefs underlying these four constructs, that best explain parent, and peer

influences on smoking in Gaborone and Francistown” in the second paragraph. The two paragraphs do not link well. I would suggest the authors use the first paragraph to mention the goal of using TPB constructs directly. Then, they may list some examples on each TPB construct. For example, the peer and parental influences are obvious subjective norms.

5. In the Materials and Methods section, the authors should have a section talking about their assessment on TPB elements. The authors are suggested reading prior TPB studies to know how to describe their TPB elements.

6. The authors mentioned that they exclude missing data. Then, they should report the missing size to let the readers understand to what extent we can trust in the findings.

7. In the Results section, I think that the authors do not need to spell out SD. SD is a commonly and widely understood statistical term. Therefore, please remove “standard deviation”.

8. Table 2 should report both frequency and percentage. Also, I cannot understand the meaning of Yes… at the top left column.

9. The authors should use a (or more) table to summarize their findings on odds ratio. Reading the text in the Results section is very easy to lose the direction. Also, P7. The sentence “Twenty-nine percent (29%) of students had tried smoking” should be changed because the authors need not to mention 29% twice.

10. The authors mentioned adjusted odds ratio in the Results section; however, I did not see the authors describe how they constructed a multivariable logistic regression model in the Data analysis section.

11. Sentences like “Eighty percent (2043) of the students” should be changed to “Eighty percent (n=2043) …” or “Two thousand and forty-three (80%) ….” because it is not intuitive to know that 2043 indicates the number.

12. In the Discussion, please add a limitation for the use of TPB. I agree that TPB is a good theory to explain many behaviors; however, it has been criticized due to its simplicity. That is, some scholars feel that human behaviors are much more complicated than the four elements (attitude, subjective norm, perceived behavioral control, and intention) proposed in the TPB. Therefore, there is a trend of using extended TPB in the realm. Specifically, scholars are encouraged to include other potential factors in the TPB to explain each specific behavior. For example, extended TPB has been used to explain the self-care behaviors among patients with diabetes by adding risk perception and health literacy on the TPB elements; some used extended TPB to explain the weight reduction behaviors by adding weight-related self-stigma. Please refer to the following references.

Lin, C.-Y., Cheung, M. K. T., Hung, A. T. F., Poon, P. K. K., Chan, S. C. C., & Chan, C. C. H. (2020). Can a Modified Theory of Planned Behavior Explain the Effects of Empowerment Education for People with Type 2 Diabetes? Therapeutic Advances in Endocrinology and Metabolism, 11, 1-12.

Cheng, O. Y., Yam, C. L. Y., Cheung, N. S., Lee, P. L. P., Ngai, M. C., & Lin, C.-Y. (2019). Extended Theory of Planned Behavior on eating and physical activity. American Journal of Health Behavior, 43(3), 569-581.

6. PLOS authors have the option to publish the peer review history of their article (what does this mean?). If published, this will include your full peer review and any attached files.

Reviewer #1: No

---

## [Author Response · Author response to Decision Letter 0]

29 Apr 2020

Academic Editor

Noted, the manuscript has been edited to meet PLOS ONE’s style requirements.

2. Please include your tables as part of your main manuscript and remove the individual files. Please note that supplementary tables (should remain/ be uploaded) as separate "supporting information" files

 Noted, the Tables have been included in the mainscript.

3. We noticed you have some minor occurrence of overlapping text with the following previous publication(s), which needs to be addressed:https://psycnet.apa.org/record/2008-17146-000

https://link.springer.com/article/10.1007/s10964-014-0187-7 In your revision ensure you cite all your sources (including your own works), and quote or rephrase any duplicated text outside the methods section. Further consideration is dependent on these concerns being addressed.

We have taken our manuscript through ant-plagiarism software and corrected were necessary.

4. Thank you for stating the following in the Acknowledgments Section of your manuscript:" This work was supported by the Bill and Melinda Gates Foundation grant OPP082662. The funders

did not have any additional role in the study design, data collection and analysis, decision to public

or preparation of the manuscript. "We note that you have provided funding information that is not currently declared in your Funding Statement. However, funding information should not appear in the Acknowledgments section or other areas of your manuscript. We will only publish funding information present in the Funding Statement section of the online submission form. Please remove any funding-related text from the manuscript and let us know how you would like to update your Funding Statement. Currently, your Funding Statement reads as follows: "The funders had no role in study design, data collection and analysis, decision to publish, or preparation of the manuscript."

Funding information has been deleted from Acknowledgement section. The funding statement must read: 

“This research was commissioned by Good Business (London) as part of a social marketing programme designed to prevent teenage girls from taking up smoking in Botswana. This was funded through a grant from the Bill and Melinda Gates Foundation Grant number OPP1082662. The funder had no role in the study design, collection, analysis and interpretation of the data; writing the report or the decision to submit the report for publication. The views expressed in this paper are therefore those of the authors and do not necessarily reflect the views of the funding body.” 

5. Thank you for stating the following in the Funding section of your manuscript: "This study was funded through the financial support of Good Business Ltd." We note that you received funding from a commercial source: Good Business Ltd. Please provide an amended Competing Interests Statement that explicitly states this commercial funder, along with any other relevant declarations relating to employment, consultancy, patents, products in development, marketed products, etc. Within this Competing Interests Statement, please confirm that this does not alter your adherence to all PLOS ONE policies on sharing data and materials by including the following statement: "This does not alter our adherence to PLOS ONE policies on sharing data and materials.” (as detailed online in our guide for authors http://journals.plos.org/plosone/s/competing-interests). If there are restrictions on sharing of data and/or materials, please state these. Please note that we cannot proceed with consideration of your article until this information has been declared. Please include your amended Competing Interests Statement within your cover letter. We will change the online submission form on your behalf.

Noted. The correct statement should be: “This research was commissioned by Good Business (London) as part of a social marketing programme designed to prevent teenage girls from taking up smoking in Botswana. This was funded through a grant from the Bill and Melinda Gates Foundation Grant number OPP1082662. The funder had no role in the study design, collection, analysis and interpretation of the data; writing the report or the decision to submit the report for publication. The views expressed in this paper are therefore those of the authors and do not necessarily reflect the views of the funding body.

”

6. Thank you for stating the following in the Competing Interests section: "The authors have declared that no competing interest exists." We note that one or more of the authors are employed by a commercial company: Good Business Ltd. Please provide an amended Funding Statement declaring this commercial affiliation, as well as a statement regarding the Role of Funders in your study. If the funding organization did not play a role in the study design, data collection and analysis, decision to publish, or preparation of the manuscript and only provided financial support in the form of authors' salaries and/or research materials, please review your statements relating to the author contributions, and ensure you have specifically and accurately indicated the role(s) that these authors had in your study. You can update author roles in the Author Contributions section of the online submission form. Please also include the following statement within your amended Funding Statement.

“The funder provided support in the form of salaries for authors [insert relevant initials], but did not have any additional role in the study design, data collection and analysis, decision to publish, or preparation of the manuscript. The specific roles of these authors are articulated in the ‘author contributions’ section. ”If your commercial affiliation did play a role in your study, please state and explain this role within your updated Funding Statement.

This research was commissioned by Good Business (London) as part of a social marketing programme designed to prevent teenage girls from taking up smoking in Botswana. 

The commercial affiliation did not play any role in the study.

7. Please also provide an updated Competing Interests Statement declaring this commercial affiliation along with any other relevant declarations relating to employment, consultancy, patents, products in development, or marketed products, etc. Within your Competing Interests Statement, please confirm that this commercial affiliation does not alter your adherence to all PLOS ONE policies on sharing data and materials by including the following statement: "This does not alter our adherence to PLOS ONE policies on sharing data and materials.” (as detailed online in our guide for authors http://journals.plos.org/plosone/s/competing-interests) . If this adherence statement is not accurate and there are restrictions on sharing of data and/or materials, please state these. Please note that we cannot proceed with consideration of your article until this information has been declared. Please include both an updated Funding Statement and Competing Interests Statement in your cover letter. We will change the online submission form on your behalf. Please know it is PLOS ONE policy for corresponding authors to declare, on behalf of all authors, all potential competing interests for the purposes of transparency. PLOS defines a competing interest as anything that interferes with, or could reasonably be perceived as interfering with, the full and objective presentation, peer review, editorial decision-making, or publication of research or non-research articles submitted to one of the journals. Competing interests can be financial or non-financial, professional, or personal. Competing interests can arise in relationship to an organization or another person. Please follow this link to our website for more details on competing interests: http://journals.plos.org/plosone/s/competing-interests

Noted

8. In your Data Availability statement, you have not specified where the minimal data set underlying the results described in your manuscript can be found. PLOS defines a study's minimal data set as the underlying data used to reach the conclusions drawn in the manuscript and any additional data required to replicate the reported study findings in their entirety. All PLOS journals require that the minimal data set be made fully available. For more information about our data policy, please see http://journals.plos.org/plosone/s/data-availability. Upon re-submitting your revised manuscript, please upload your study’s minimal underlying data set as either Supporting Information files or to a stable, public repository and include the relevant URLs, DOIs, or accession numbers within your revised cover letter. For a list of acceptable repositories, please see http://journals.plos.org/plosone/s/data-availability#loc-recommended-repositories. Any potentially identifying patient information must be fully anonymized. Important: If there are ethical or legal restrictions to sharing your data publicly, please explain these restrictions in detail. Please see our guidelines for more information on what we consider unacceptable restrictions to publicly sharing data: http://journals.plos.org/plosone/s/data-availability#loc-unacceptable-data-access-restrictions. Note that it is not acceptable for the authors to be the sole named individuals responsible for ensuring data access. We will update your Data Availability statement to reflect the information you provide in your cover letter.

Noted, the data has been attached as supporting information file.

9. Please include a separate caption for each figure in your manuscript.

Noted, we have included the separate caption.

10. Please ensure that you refer to Figure 2 in your text as, if accepted, production will need this reference to link the reader to the figure.

Noted.

Comments to the Author

1. Is the manuscript technically sound, and do the data support the conclusions?

Reviewer #1: No

2. Has the statistical analysis been performed appropriately and rigorously? 

Reviewer #1: I Don't Know

3. Have the authors made all data underlying the findings in their manuscript fully available?

Reviewer #1: Yes

4. Is the manuscript presented in an intelligible fashion and written in standard English?

Reviewer #1: No

5. Review Comments to the Author

Reviewer #1: The manuscript entitled “The theory of planned behavior as a behavior change model for tobacco control strategies among adolescents in Botswana” recruited 2554 adolescents to study whether the theory of planned behavior (TPB) can be a useful theory to examine how far we can understand the behavior of smoking among adolescents inhabiting in Botswana. I applaud for the authors to collect such valuable data on assessing a widely used theory to study a meaningful and important research question. Specifically, the present study has a strong strength in data collection; that is, the authors used a rigorous sampling method to collect a large sample. However, I have following comments for the authors to elevate their work. In addition, I believe that the manuscript needs to be edited by a native English speaker to make it free from grammatical errors (e.g., P5. Our analyses focuses on…)

The manuscript has been edited by Native English speakers.

1. A major and root problem is that the authors do not clearly indicate whether they studied on the smoking behavior. In many descriptions, the authors only talked about smoking intention (e.g., in the Abstract). However, the authors have collected whether the participants are a current smoker and they sometimes reveal the relationship between the TPB elements and a current smoker or not. Therefore, this causes a lot of confusion when I read the manuscript. I cannot know what the authors stand in using the TPB.

This has been clearly indicated in the revised version. The prevalence of smoking was calculated in a previous manuscript from the same data and the prevalence has been cited in the current paper. Thus the authors looked at current smoking and intentions to smoke.

2. In the Abstract, the sentence in the Background is unclear. Specifically, it is unclear whose intentions, whose attitudes, whose subjective norms, and whose perceptions that promote/discourage the smoking behavior of an individual. Also, do authors want to say perceived behavioral control when they mention “people’s perceptions”? If yes, then, the authors should not use people’s perceptions to indicate perceived behavioral control; they are different concepts. Following this, the Objective is not in line with what the authors did in the study. Specifically, the authors studied more than “behavioral attitudes” and “intention”, while the authors only mentioned the two factors in the Objective.

We edited the statement to refer to adolescents. “Behavioral intentions (motivational factors), attitudes, subjective norm (social pressures), and perceived behavioral control promote or discourage smoking behavior among adolescents.”

We edited to behavioral control not perceptions

We edited the objective to include all the constructs of TPB.

In this study, we utilized TPB to assess attitudes, subjective norms, perceived behavioral control and intentions of adolescents on smoking in Gaborone and Francistown, the two largest cities of Botswana.

3. In the Introduction, the authors said, “The TPB has also been used successfully to predict and explain a wide range of health behaviors including exercise, smoking and drug use, HIV prevention behaviors, among others (Montaño & Kasprzyk, 2002).” I agree. However, please cite more references to support this sentence. Please refer to the following.

Hou, W.-L., Lin, C.-Y., Wang, Y.-M., Tseng, Y.-H., & Shu, B.-C. (2020). Assessing Related Factors of Intention to Perpetrate Dating Violence among University Students Using the Theory of Planned Behavior. International Journal of Environmental Research and Public Health, 17(3), 923.

Fung, X. C. C., Pakpour, A. H., Wu, K.-Y., Fan, C.-W., Lin, C.-Y., Tsang, H. H. W. (2019). Psychosocial variables related to weight-related self-stigma in physical activity among young adults across weight status. International Journal of Environmental Research and Public Health, 17, 64.

Lin, C.-Y., Broström, A., Årestedt, K., Mårtensson, J., Steinke, E. E., & Pakpour, A. H. (2020). Using extended Theory of Planned Behavior to determine factors associated with help-seeking behavior of sexual problems in women with heart failure: A longitudinal study. Journal of Psychosomatic Obstetrics & Gynecology, 41,54-61.

Lin, C.-Y., Broström, A., Nilsen, P., & Pakpour, A. H. (2018). Using extended Theory of Planned Behavior to understand aspirin adherence in pregnant women. Pregnancy Hypertension: An International Journal of Women’s Cardiovascular Health, 12, 84-89.

Lin, C.-Y., Fung, X. C. C., Nikoobakht, M., Burri, A., & Pakpour, A. H. (2017). Using theory of planned behavior incorporated with perceived barriers to explore sexual counseling services delivered by health professionals in individuals suffering from epilepsy. Epilepsy & Behavior, 74, 124-129.

Strong, C., Lin, C.-Y., Jalilolghadr, S., Updegraff, J. A., Broström, A., & Pakpour, A. H. (2018). Sleep hygiene behaviors in Iranian adolescents: an application of the Theory of Planned Behavior. Journal of Sleep Research, 27(1), 23-31.

Lin, C.-Y., Oveisi, S., Burri, A., & Pakpour, A. H. (2017). Theory of Planned Behavior including self-stigma and perceived barriers explain help-seeking behavior for sexual problems in Iranian women suffering from epilepsy. Epilepsy & Behavior, 68, 123-128.

Lin, C.-Y., Updegraff, J. A., & Pakpour, A. H. (2016). The relationship between the theory of planned behavior and medication adherence in patients with epilepsy. Epilepsy & Behavior, 61, 231-236.

We have cited all the references provided by the reviewer. 

The TPB has also been used successfully to predict and explain a wide range of health behaviors including exercise, smoking and drug use, HIV prevention behaviors, among others (Fung et al., 2020; Hou, Lin, Wang, Tseng, & Shu, 2020; Lin et al., 2019; Lin, Broström, Nilsen, & Pakpour, 2018; Lin, Updegraff, & Pakpour, 2016; Lin, Fung, Nikoobakht, Burri, & Pakpour, 2017; Lin, Oveisi, Burri, & Pakpour, 2017; Montaño & Kasprzyk, 2002; Strong et al., 2018).

4. The Goals of the study section in the Introduction should be rewritten. Specifically, I cannot see the link between the two paragraphs: the authors said “The current study assessed the effect of peer and parental influences on youth smoking” in the first paragraph and “Our analyses focuses on the four belief-based TPB constructs (attitudes, subjective norms, perceived behavioral control and intentions) because these are most conducive to change with persuasive messaging in communication campaigns. The main objective of our study was to identify the key beliefs underlying these four constructs, that best explain parent, and peer influences on smoking in Gaborone and Francistown” in the second paragraph. The two paragraphs do not link well. I would suggest the authors use the first paragraph to mention the goal of using TPB constructs directly. Then, they may list some examples on each TPB construct. For example, the peer and parental influences are obvious subjective norms.

We used the first paragraph to mention the goal of using TPB constructs directly and went on to list some examples on each TPB construct. 

5. In the Materials and Methods section, the authors should have a section talking about their assessment on TPB elements. The authors are suggested reading prior TPB studies to know how to describe their TPB elements.

We added a section on main exposures of interest. The section describes assessments of TPB elements.

6. The authors mentioned that they exclude missing data. Then, they should report the missing size to let the readers understand to what extent we can trust in the findings.

We removed the statement of missing data.

7. In the Results section, I think that the authors do not need to spell out SD. SD is a commonly and widely understood statistical term. Therefore, please remove “standard deviation”.

We removed SD in Results section.

8. Table 2 should report both frequency and percentage. Also, I cannot understand the meaning of Yes… at the top left column.

We edited the table to include both frequency and percentage. We removed Yes at the top left column. It’s now Table 3 not 2.

9. The authors should use a (or more) table to summarize their findings on odds ratio. Reading the text in the Results section is very easy to lose the direction. Also, P7. The sentence “Twenty-nine percent (29%) of students had tried smoking” should be changed because the authors need not to mention 29% twice.

Table 2 has been added to summarise findings on odds ratio

We edited the sentence to a more accurate sentence as follows: “Twenty-nine percent of students had tried smoking.”

10. The authors mentioned adjusted odds ratio in the Results section; however, I did not see the authors describe how they constructed a multivariable logistic regression model in the Data analysis section.

We added more information in data analysis section on logistic regression: “Logistic regression was used to determine adjusted odds ratios (AOR) and their 95% CIs, for independent variables (attitudes, subjective norms, perceived behavioral control and intentions) associated with active smoking. For the main analyses, smoking was categorized as a dichotomic variable (yes vs. no). Secondary analyses using linear regression was used to determine the association when smoking was treated as a continuous variable in terms of the number of cigarettes smoked per month. A p-value of less than 0.05 was considered to indicate statistical significance.” 

11. Sentences like “Eighty percent (2043) of the students” should be changed to “Eighty percent (n=2043) …” or “Two thousand and forty-three (80%) ….” because it is not intuitive to know that 2043 indicates the number.

We have edited the sentences according to reviewer’s comments.

“Eighty percent (n = 2043) of the students felt they could refuse a cigarette if a friend offered”.

12. In the Discussion, please add a limitation for the use of TPB. I agree that TPB is a good theory to explain many behaviors; however, it has been criticized due to its simplicity. That is, some scholars feel that human behaviors are much more complicated than the four elements (attitude, subjective norm, perceived behavioral control, and intention) proposed in the TPB. Therefore, there is a trend of using extended TPB in the realm. Specifically, scholars are encouraged to include other potential factors in the TPB to explain each specific behavior. For example, extended TPB has been used to explain the self-care behaviors among patients with diabetes by adding risk perception and health literacy on the TPB elements; some used extended TPB to explain the weight reduction behaviors by adding weight-related self-stigma. Please refer to the following references.

Lin, C.-Y., Cheung, M. K. T., Hung, A. T. F., Poon, P. K. K., Chan, S. C. C., & Chan, C. C. H. (2020). Can a Modified Theory of Planned Behavior Explain the Effects of Empowerment Education for People with Type 2 Diabetes? Therapeutic Advances in Endocrinology and Metabolism, 11, 1-12.

Cheng, O. Y., Yam, C. L. Y., Cheung, N. S., Lee, P. L. P., Ngai, M. C., & Lin, C.-Y. (2019). Extended Theory of Planned Behavior on eating and physical activity. American Journal of Health Behavior, 43(3), 569-581.

We have incorporated the limitations of the theory of planned behaviour.

Future studies could extend the TBP theory by adding risk perceptions and healthy literacy with regards smoking among adolescents.

6. PLOS authors have the option to publish the peer review history of their article (what does this mean?). If published, this will include your full peer review and any attached files.

Do you want your identity to be public for this peer review? For information about this choice, including consent withdrawal, please see our Privacy Policy.

Reviewer #1: No

---

## [Decision Letter · Decision Letter 1]

1 May 2020

PONE-D-20-00392R1

The theory of planned behavior as a behavior change model for tobacco control strategies among adolescents in Botswana.

PLOS ONE

Dear Dr Tapera,

Thank you for submitting your manuscript to PLOS ONE. After careful consideration, we feel that it has merit but does not fully meet PLOS ONE’s publication criteria as it currently stands. Therefore, we invite you to submit a revised version of the manuscript that addresses the points raised during the review process.

We would appreciate receiving your revised manuscript by Jun 15 2020 11:59PM. To enhance the reproducibility of your results, we recommend that if applicable you deposit your laboratory protocols in protocols.io, where a protocol can be assigned its own identifier (DOI) such that it can be cited independently in the future. For instructions see: http://journals.plos.org/plosone/s/submission-guidelines#loc-laboratory-protocols

We look forward to receiving your revised manuscript.

Kind regards,

Amir H. Pakpour, Ph.D.

Academic Editor

PLOS ONE

Reviewers' comments:

Reviewer's Responses to Questions

**Comments to the Author**

1. If the authors have adequately addressed your comments raised in a previous round of review and you feel that this manuscript is now acceptable for publication, you may indicate that here to bypass the “Comments to the Author” section, enter your conflict of interest statement in the “Confidential to Editor” section, and submit your "Accept" recommendation.

Reviewer #1: All comments have been addressed

2. Is the manuscript technically sound, and do the data support the conclusions?

Reviewer #1: Yes

3. Has the statistical analysis been performed appropriately and rigorously? 

Reviewer #1: Yes

4. Have the authors made all data underlying the findings in their manuscript fully available?

Reviewer #1: Yes

5. Is the manuscript presented in an intelligible fashion and written in standard English?

Reviewer #1: Yes

6. Review Comments to the Author

Reviewer #1: The revised manuscript is much improved, especially the English presentation. The authors also have answered all my prior comments in a satisfactory way. However, one minor issue needs to be fixed before acceptance. That is, the manuscript contains both British and American spelling (e.g., utilise; behavioral; behavioural). Please use Word function to check whether all the spellings are consistent and correct them in either British spelling or American spelling. Please do not use a mixture. Anyway, good work and I look forward to reviewing a new revision.

7. PLOS authors have the option to publish the peer review history of their article (what does this mean?). If published, this will include your full peer review and any attached files.

Reviewer #1: No

---

## [Author Response · Author response to Decision Letter 1]

2 May 2020

We have edited the spellings to American English for example on page 4 we have changed the word behaviour to behavior and on page 5 we changed the word utilised to utilized. We have gone through the whole document using Word function to check for consistence in spellings.

---

## [Editor Report · Decision Letter 2]

6 May 2020

The theory of planned behavior as a behavior change model for tobacco control strategies among adolescents in Botswana.

PONE-D-20-00392R2

Dear Dr. Tapera,

We are pleased to inform you that your manuscript has been judged scientifically suitable for publication and will be formally accepted for publication once it complies with all outstanding technical requirements.

With kind regards,

Amir H. Pakpour, Ph.D.

Academic Editor

PLOS ONE
---

## [Editor Report · Acceptance letter]

28 May 2020

PONE-D-20-00392R2 

The theory of planned behavior as a behavior change model for tobacco control strategies among adolescents in Botswana. 

Dear Dr. Tapera:

I am pleased to inform you that your manuscript has been deemed suitable for publication in PLOS ONE. Congratulations! Your manuscript is now with our production department. 

With kind regards,

on behalf of

Dr. Amir H. Pakpour 

Academic Editor

PLOS ONE